# Recent Advances in Natural Deastringency and Genetic Improvement of Chinese PCNA Persimmon (*Diospyros kaki*)

Sichao Yang [1,*], Meng Zhang [1], Lei Xu [1,*], Qinglin Zhang [2], Chaohua Zhou [1], Xinlong Hu [1] and Zhengrong Luo [2]

1   Horticultural Institute, Jiangxi Academy of Agricultural Sciences, Nanchang 330200, China; zhangmeng2015430@163.com (M.Z.); zch8036@163.com (C.Z.); h15870008656@126.com (X.H.)
2   National Key Laboratory for Germplasm Innovation & Utilization of Horticultural Crops, College of Horticulture and Forestry Sciences, Huazhong Agricultural University, Wuhan 430070, China; luozhr@mail.hzau.edu.cn (Z.L.)
*   Correspondence: yangsichao1991@163.com (S.Y.); xulei_1209@163.com (L.X.)

**Abstract:** Persimmon (*Diospyros kaki*) is a worldwide fruit cultivated mainly in the East Asia, Mediterranean, Caucasus, Latin America, and Oceania regions. This fruit contains abundant proanthocyanidins (PAs, also called condensed tannins), whose biosynthesis is the main cause of fruit astringency. As the original centre and top producing country, China has discovered a unique type with desirable natural deastringency, the Chinese pollination-constant non-astringent (C-PCNA) persimmon. Studies have revealed that the C-PCNA trait is controlled by a single and dominant locus, which differs from that of another type, the Japanese PCNA type, with recessive loci. In the C-PCNA type, accumulating evidence has shown that the astringency removal process involves two pathways ("dilution effect" and "coagulation effect"). Moreover, molecular marker-assisted selection (MAS) for the natural deastringency trait locus in C-PCNA has been used to test the non-astringency/astringency trait of hybrid offspring at the seedling stage. Importantly, persimmon can bear male flowers, female flowers, and perfect flowers, but sex-linked MAS has been developed for female-only persimmon. This sex-linked MAS, together with astringency-linked MAS and embryo rescue technology, may even shorten the conventional cross-breeding period of about 2–3 years. In addition, recently studies have established a stable genetic transformation system for persimmon transgenic improvement. Despite these efforts, how synthetic PAs and metabolism pathways lead to a deastringent trait remains unclear for persimmon. Thus, our review summarizes the latest research progress on the natural deastringency mechanism in C-PCNA, and we provide a new viewpoint for the genetic improvement of persimmon breeding in China.

**Keywords:** *Diospyros kaki*; proanthocyanidin; natural deastringency; marker-assisted selection; C-PCNA breeding

## 1. Introduction

Persimmon (*Diospyros kaki* Thunb., 2n = 6x = 90, fewer varieties 2n = 9x = 135) originates from China, where the main producing area of persimmon began to develop from the traditional yellow river basin to the Yangtze River basin and its southern region. With the Yangtze River as the boundary, the top 10 provinces in terms of annual output account for 50% each (Figure 1) [1]. In recent years, in addition to the traditional production areas of China, Korea, Japan, and Brazil, other countries, such as Spain, Azerbaijan, Uzbekistan, Australia, and Italy, have also begun to develop the persimmon industry on a large scale [1]. Persimmon is categorized based on whether the fruit loses its astringency on the tree (genetic trait of natural deastringency) into the pollination-constant non-astringency (PCNA) type, which is controlled by a single locus, and the non-PCNA-type, which is controlled by a quantitative trait locus. PCNA persimmons include the Chinese PCNA (C-PCNA) and Japanese PCNA (J-PCNA) cultivars (Figure 2A). Whether from a genetic relationship,

the genetic characteristics of non-astringency and astringency traits, or the accumulation pattern and composition of tannins, the natural deastringency mechanisms of C-PCNA and J-PCNA are obviously different. The C-PCNA trait of astringency loss is dominant (*CPCNA*) over non-PCNA types, while the J-PCNA is recessive (*AST/ast*), controlled by a different allele. In addition, the non-PCNA type includes the pollination-variant non-astringent (PVNA), pollination-variant astringent (PVA), and pollination-constant astringent (PCA) types [2–5]. Therefore, only J-PCNA hybridization could obtain a PCNA in the $F_1$ generation; if J-PCNA was crossed with non-PCNA, there would be no PCNA individual in the $F_1$ generation, and even if it was backcrossed with PCNA, the proportion of PCNA in the $F_2$ generation would be only 15~20%. Nevertheless, whether C-PCNA was crossed with J-PCNA or with non-PCNA, 50% C-PCNA would be obtained in the $F_1$ generation [6] (Table 1).

**Table 1.** Persimmon classification and characteristics.

| Classification | Representative Varieties | Origin Area | Deastringency Method | Deastringency Period | Genetic Characteristics | Key Transcript Factor |
|---|---|---|---|---|---|---|
| C-PCNA | Eshi 1 Luotian Tianshi | China | Dilution effect and coagulation effect | 25 WAB | Quality traits/ Dominant [6] | *DkMYB14* [7] |
| J-PCNA | Youhou | Japan | Dilution effect | 15 WAB | Quality traits/ Recessive [6] | *DkMYB4* [8] |
| PVNA | Huangjin Fangshi | China | Partially deastringent | Not fully natural deastringency | Quantitative traits [6] | - |
| PCA | Huashi 1 | China | Partially deastringent | Not fully natural deastringency | Quantitative traits [6] | - |
| PVA | Mopanshi Gongcheng Shuishi | China | Not natural deastringency | Not natural deastringency | Quantitative traits [6] | - |

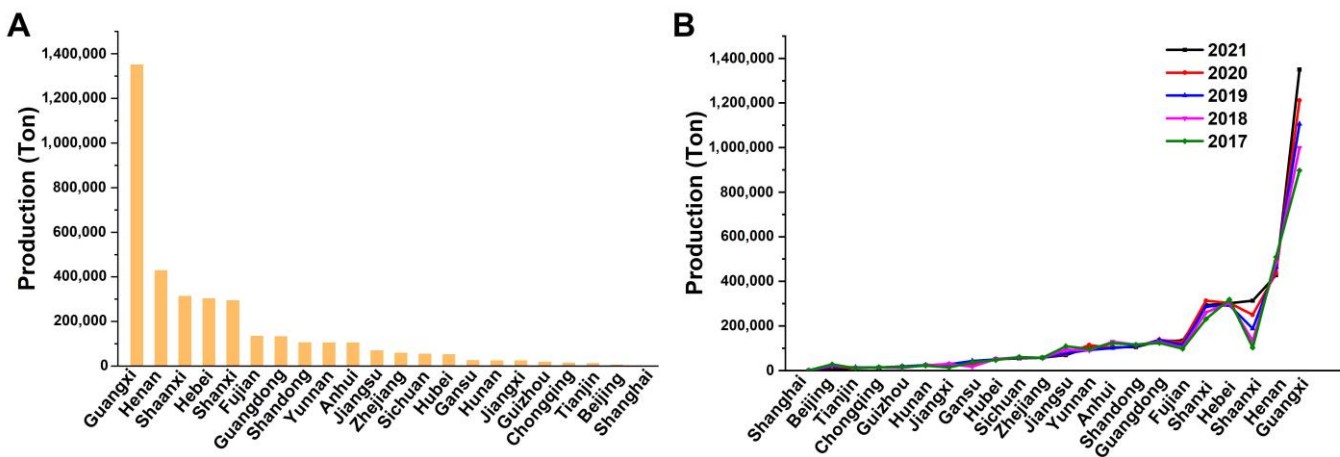

**Figure 1.** Persimmon production (tons) in producing provinces in mainland China in 2021 (**A**) and changes in persimmon production (tons) in producing provinces in mainland China from 2017 to 2021 (**B**).

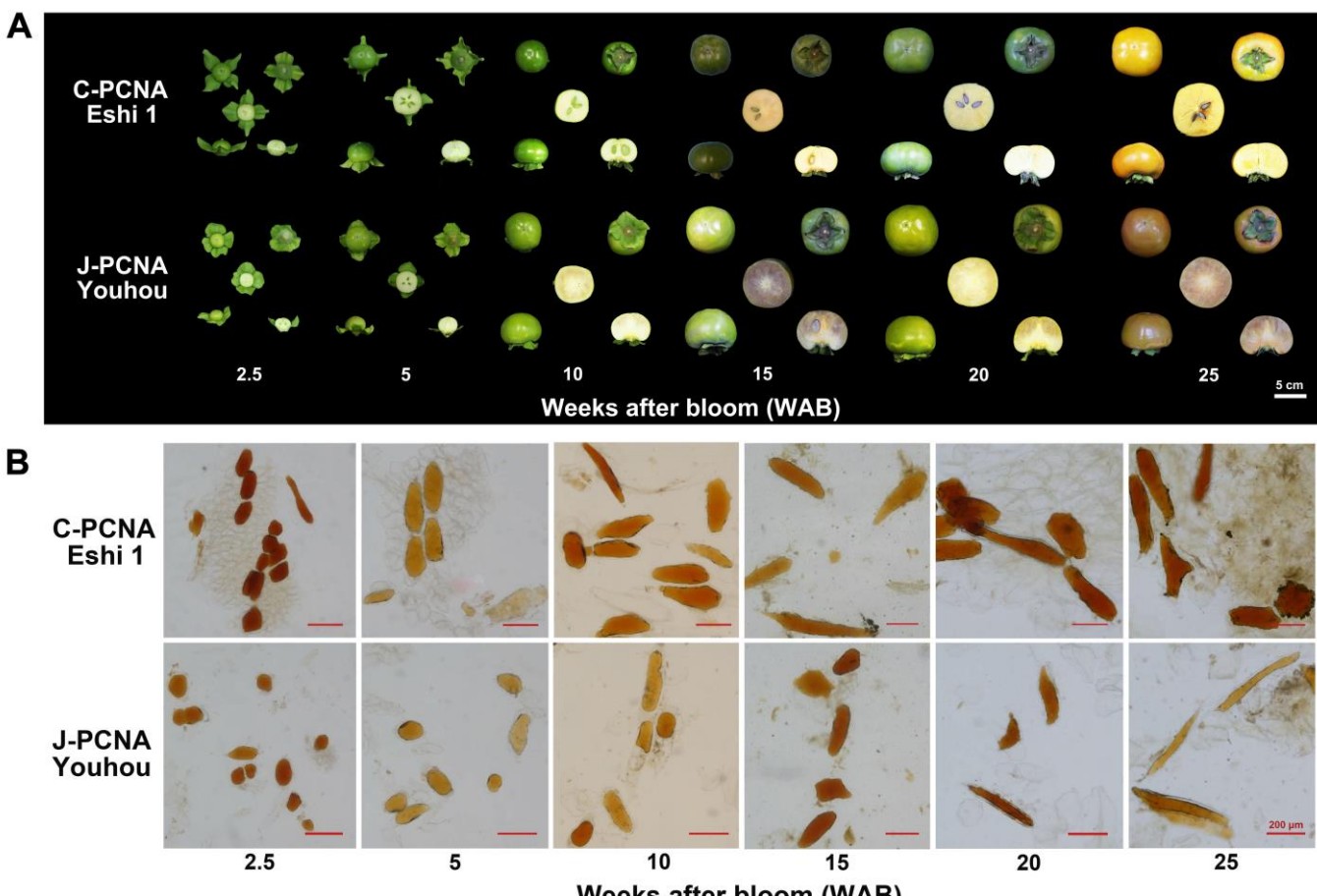

**Figure 2.** Fruit (**A**) and tannin cell (**B**) morphological characteristics of C-PCNA and J-PCNA during fruit development.

Plant tannins, also called vegetable tannins, which belong to secondary metabolites, usually refer to the complex polyphenols with a molecular weight in the range of 500–3000 Da [9]. Based on the chemical structure characteristics of tannins, they can be divided into two categories: hydrolysable tannins and condensed tannins [10]. Hydrolysed tannins are ester substances formed by gallic acid and its related compounds, which belongs to C6·C1 phenolic acid. Tannins in persimmon are mainly condensed tannins, which accumulate in the vacuoles of specific cells, called tannin cells [3]. These metabolites play an important role in protecting the plants. For example, proanthocyanidins (PAs) prevent the erosion of seeds caused by fungi in feed plants [11], and the presence of PAs is considered an important characteristic of feed crops to prevent the bloating caused by grazing and to improve the nitrogen nutrition of ruminant livestock [12]. Its strong antioxidant capacity is also very beneficial to human health by effectively preventing cancer and cardiovascular diseases [13–16], relieving hypercholesterolaemia, as well as functioning in antioxidation and scavenging free radicals [17–19]. Moreover, persimmon tannins are also widely used as a raw material or additive for everyday products, chemical products, and health care products [20,21]. However, PAs can also precipitate gelatine, alkaloids, and other proteins, especially saliva proteins in the mouth, resulting in the astringent characteristics of foods with a high PAs content [22].

According to whether they dissolve in alcohol solutions, condensed tannins can be divided into soluble tannins and insoluble tannins [23]. Soluble tannins can coagulate with oral proteins, resulting in a dry or puckering sensation. When the soluble tannin content in the fruit decreases to less than 0.1%, the PCNA fruit completes the natural deastringency

process [24–26]. As PCNA fruits can be edible without any other artificial treatment, this type of variety is thus crucial for persimmon breeding and genetic improvement [1].

The core parents of J-PCNA originated from a narrow region centred on the Kinki and Tokai regions of Japan with a high genetic similarity. Thus, the size, yield, and growth potential of the offspring exhibited significant degradation due to long-term inbreeding [27]. The natural deastringency process of J-PCNA is primarily controlled by the *DkbZIP5* and *DkMYB4* transcription factors that stop expressing at the early stage of fruit development, which leads to the significant downregulation of the structural genes involved in the biosynthesis of tannins [8,28]. As the fruit expands, the tannin content sharply decreases to less than 0.1% at 10 weeks after bloom [27]. The C-PCNA varieties are concentrated in the Dabie Mountains at the junction of the Hubei, Henan, and Anhui Provinces in China, of which the quantities of PCNA resources were the highest in Luotian County, Hubei Province, where the first C-PCNA variety, 'Luotian Tianshi', was discovered in 1983 [29,30]. At present, the natural deastringency mechanism of C-PCNA is related to both the "dilution effect" and "coagulation effect", and it is more biased towards the latter [31]. The natural deastringency process in C-PCNA is more complicated than that in J-PCNA, and it may be regulated by multilevel and multinode regulatory networks. Based on revelation of the genetic background and regulatory networks of the natural deastringency trait, molecular markers, embryo culture, and genetic transformation have been used to accelerate PCNA breeding. This review summarizes the latest research progress on tannin metabolic characteristics, natural deastringency mechanisms, PCNA breeding, and transgenic breeding, aiming to provide a theoretical basis and breakthrough for persimmon research.

## 2. Tannin Composition and Content Change in C-PCNA

### 2.1. Tannin Cell Characteristics in C-PCNA

Specific cells, called tannin cells, exist in C-PCNA fruit, and the size, shape, and composition of these cells change during fruit development [32–34]. According to their longitudinal to transverse diameter ratio and geometric shape, tannin cells can be divided into six types: slender, prolate, ellipsoid, subcircular, polygonal, and cuspidate. According to their surface morphology, tannin cells can be divided into four types: acanthoid, tuberculiform, pitted, and naturally smooth [34–37]. The developmental dynamics of tannin cells in C-PCNA and J-PCNA are not the same; for example, except for the coagulated browning tannin cells, there are a few dark reddish brown condensed tannin cells in J-PCNA but not in C-PCNA [24] (Figure 2B). The tannin cells in C-PCNA 'Luotian Tianshi' and 'Eshi 1' are usually prolate [37]. In general, the tannin cells in PCNA are smaller than those in non-PCNA, and the density of tannin cells in PCNA is also lower than that of tannin cells in non-PCNA [38,39].

### 2.2. Tannin Composition in C-PCNA

In 1923, the astringent phenolic substance was first extracted from the juice of unripe astringent persimmon by Komatsu and Mutsunami. Its molecular structure was studied and the molecular formula was initially confirmed as $C_{14}H_{20}O_9$. Further studies showed that persimmon tannins may contain gallic acid and m-benzene [17]. Ito and Oshima (1962) suggested that persimmon tannins are not only colourless lupidin, but also contained more complex components. When persimmon tannins were degraded by strong acid thermal degradation, not only a large amount of colourless cynaridine was produced, but also acids, catechins (C), and catechin acids were produced [40]. By using the benzyl mercaptan degradation method, Matsuo and Ito (1978) analysed the tannin structure of the pollination-variant astringent 'Hiratanenashi' fruit and found that the molecular weight of the tannins was $1.38 \times 10^4$ Da, which mainly contain catechin, catechin-3-gallate (CG), gallocatechin (GC), and gallocatechin 3-gallate (GCG), i.e., four monomer compounds [17,41,42]. These four monomers are connected to each other by the fourth, sixth, or eighth carbon atom to form a number ratio of 1:1:2:2 repeating units and by further polymerization to form a

polymer. Fei et al. (1999) analysed the seasonal variation in tannins in different deastringent types of persimmon, the relationship between tannins and non-astringency/astringency traits, the content of soluble tannins and the characteristics of tannins cells [33]. Their results showed that the PCNA varieties had a higher catechin content in the early stage of fruit development, and the decrease rate was lower than that of non-PCNA persimmon. Non-PCNA persimmon had a higher level of gallic acid. Compared with J-PCNA, C-PCNA 'Luotian Tianshi' had a higher content of catechin and gallic acid.

### 2.3. Changes in Tannin Contents in C-PCNA

After the qualitative analysis of the tannin contents of C-PCNA fruit during development utilizing the blotting method, the staining of the filter paper soaked in a $FeCl_2$ solution became lighter during fruit development, and this change in staining was most significant at 15–20 weeks after bloom (WAB); the staining of the filter paper was almost not evident at 25 WAB [7,43]. After quantitative analysis of the tannin contents by the Folin–Ciocalteu method, the soluble tannin content gradually decreased during fruit development, which was consistent with the results of the blotting method, and it was lower than 0.2% at 25 WAB, when C-PCNA completed the natural deastringency process. Moreover, the insoluble tannin content also showed a decreasing trend, except for an increase from 15 to 20 WAB, which indicates that there is conversion of soluble tannins to insoluble tannins involved in the natural deastringency process in C-PCNA [43].

## 3. Natural Deastringency Mechanism in C-PCNA

### 3.1. Acetaldehyde Metabolism Is the Essential Pathway Involved in the Natural Deastringency of C-PCNA

Acetaldehyde is a small-molecule substance that can directly cross the cell membrane without transporters. Plant cells produce acetaldehyde by anaerobic respiration under anaerobic conditions. Alcohol dehydrogenase (ADH), pyruvate decarboxylase (PDC), and acetaldehyde dehydrogenase (ALDH) are the three direct key enzymes involved in acetaldehyde metabolism [43]. PDC converts pyruvate into acetaldehyde, ALDH converts acetaldehyde into acetate, and ADH is involved in the reversible interconversion of ethanol and acetaldehyde. Moreover, pyruvate kinase (PK) irreversibly catalyses phosphoenolpyruvate (PEP) and ADP to form pyruvate and ATP in the glycolysis pathway.

Acetaldehyde plays an important role in the conversion of soluble tannins into insoluble tannins. Acetaldehyde methylene (-$CH_2$-) and flavan-3-ol monomers polymerize to form a gel-like substance, which results in the natural deastringency of C-PCNA [44]. At the early development stage of C-PCNA, the acetaldehyde content in the pulp decreased continuously and then remained unchanged from 15 to 20 WAB. However, it slightly increased from 20 to 25 WAB, even though it was constantly consumed to react with the soluble tannins, which indicates that acetaldehyde is heavily synthesized in the fruit at the developmental stage. After analysing the expression levels of *ADH* and *PDC* in the different tissues of C-PCNA, the results revealed that acetaldehyde may be primarily synthesized in the seeds. As predicted, the acetaldehyde content in the seeds increased constantly during fruit development and accumulated rapidly, from 15 to 25 WAB, which is consistent with the significant decreasing trend in soluble tannins at the later developmental stage in C-PCNA [43].

### 3.2. Regulation Network Is Involved in the Natural Deastringency of C-PCNA

The PAs biosynthesis pathway includes the common phenylpropane pathway, core flavonoid-anthocyanin pathway, and proanthocyanidin-specific pathway (Figure 3). The proanthocyanidin precursor, flavan-3-alcohol, is biosynthesised in the cytoplasm; then, it is transported to the vacuoles through the transmembrane transporter and is polymerized to form PAs. As it is known that the C-PCNA's loss of astringency includes two processes, the "dilution effect" and "coagulation effect", the structural genes involved in both processes directly synthesise the enzyme that participates in the biosynthesis of PAs or acetaldehyde.

For example, the relative expression levels of *F3′5′H* to *F3′H*, and *ANR* to *LAR*, were considerably higher, and the PAs composition corresponded to the seasonal expression balances in both types; these results suggest that the expressions of *F3′5′H* and *ANR* are important for PAs accumulation in persimmon fruit [45]. The full-length cDNA of *DkLAR* gene was 1356 bp long and encoded an open reading frame of 349 amino acid residues. The obtained *DkLAR* protein was closely related to the homolog in other plant species, and the expression of the *DkLAR* gene in the Chinese pollination-constant non-astringent (PCNA) genotype coincided with tannin cell development, but not in the Japanese PCNA and Chinese pollination-variant astringent (PVA) genotypes. Moreover, the *DkLAC1* gene was isolated by the homology-based clone method. DkLAC1, the predicted protein product of this gene, is a plant laccase considered to be potentially involved in PAs polymerization in C-PCNA during normal ripening. And this enzyme is phylogenetically related to the known enzyme AtLAC15, which is involved in the polymerization of PAs [43]. In addition, the allele of *DkDTX5/MATE5*-lacking Ser-84 was identified as a dominantly expressed gene in the A-type and lost its transport function; the site-directed mutagenesis revealed that DkDTX5/MATE5 binds to PAs precursors via Ser-84 [46].

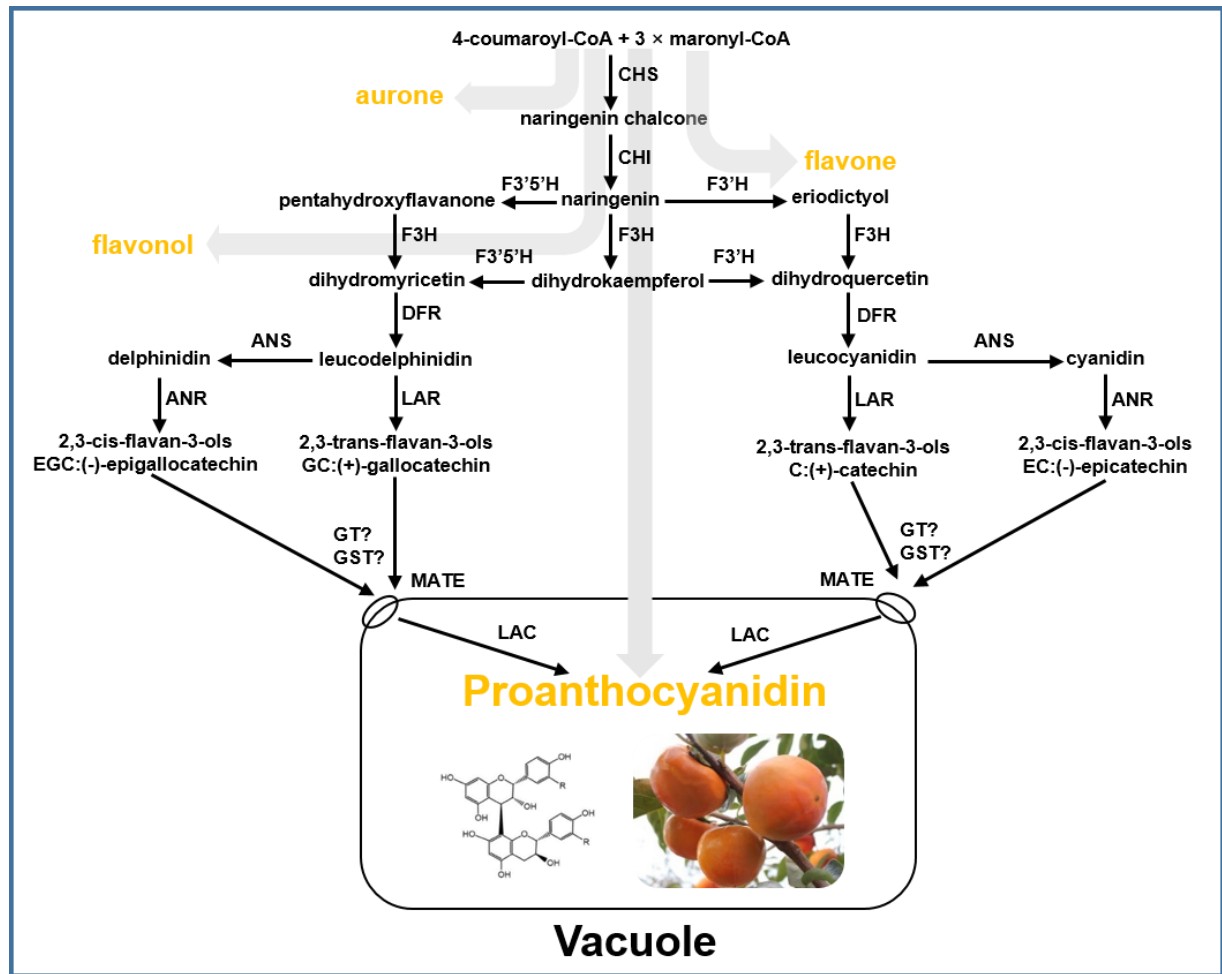

**Figure 3.** Biosynthesis pathway of proanthocyanidins. CHS, chalcone synthase; CHI, chalcone isomerase; F3H, flavanone 3 hydroxylase; F3′H; flavonoid−3′−hydroxylase; F3′5′H, flavonoid 3′5′ −hydroxylase; DFR, dihydroflavonol 4−reductase; ANS, anthocyanidin synthase; LAR, leucoanthocyanidin reductase; ANR, anthocyanidin reductase; GT, glycosyltransferase; GST, glutathione S−transferase; LAC, laccase; and MATE transporter, multidrug and toxic compound extrusion transporter.

To reveal the key factors involved in the natural deastringency of C-PCNA, homology-based cloning, high-throughput sequencing, and proteomic analysis were utilized, and many transcription factors and miRNAs were identified. A gene homologous to *DkMYB4* was isolated from C-PCNA, named *DkPA1*, which regulates *DkLAR* in both C-PCNA and non-PCNA and plays a more important role in C-PCNA than in J-PCNA [2]. Moreover, DkMYC1 was cloned and characterized in C-PCNA, which may be an important bHLH (basic helix–loop–helix) gene involved in PAs biosynthesis in persimmon [2]. Through the combined analysis of transcriptome sequencing and proteome sequencing in C-PCNA fruit at 10 and 20 WAB, as well as the samples (10 WAB) being treated with warm water for 12 h or at room temperature for 12 h, two MYB genes were screened, named *DkMYB14* and *DkMYB15*. The functional analysis verified that *DkMYB14* can act not only as a transcriptional repressor that reduces soluble PAs content by repressing the expression of *DkF3′5′H* and *DkANR*, but also as a transcriptional activator that promotes the insolubilisation of soluble PAs by directly upregulating *DkADH1* and *DkPDC2* [7]. Similarly, *DkMYB15*, also verified as a repressor, interacts with *DkbHLH2* to repress the expression of the PAs pathway genes, which results in a decrease in the PAs content [47]. The bioinformatics analysis of the transcriptome data of C-PCNA, J-PCNA, and non-PCNA at 2.5, 10, 20, and 25 WAB identified the *DkMYB21* gene, which interacts with the bHLH-type transcription factor *DkMYC1* and activates the expression of the structural genes *DkPAL*, *DkF3′5′H*, *DkDFR*, *DkANS*, and *DkANR* and the transcription factor *DkMYB4* to regulate the synthesis of PAs [48]. In addition, two WRKY transcription factors, DkWRKY3/15, could be upstream of *DkPK1* and positively regulate the natural deastringency of C-PCNA persimmon [2]. After the high-throughput sequencing of small RNA in C-PCNA fruit at 15 and 20 WAB, many miRNAs involved in natural deastringency were screened [49], among which miR858b was verified to repress *DkMYB19* and *DkMYB20* and decrease the biosynthesis of soluble tannins, and *DkMYB20* contributes to the regulation of *DkANR* to control PAs biosynthesis [2]. MiR143c represses the expression of *DkALDH10* and positively regulates the natural deastringency process of C-PCNA persimmon [50]. DkmiR397 represses the expression of *DkLAC2* and inhibits the biosynthesis of soluble tannins [51].

## 4. Conventional Cross-Breeding of C-PCNA

### 4.1. Sex Classification of Chinese Persimmon Flowers

Persimmon flower sex classification is complex, and individual persimmon floral phenotypes can be generally divided into four types: gynoecious (bearing female flowers only; most cultivars), androecious (bearing male flowers only; a few cultivars), monoecious (bearing both male and female flowers; a few cultivars), and trimonoecious (bearing both bisexual and unisexual flowers; rare cultivars) [52]. Female flowers are solitary, while male flowers and complete flowers are cymes. For females, usually there is only one well-developed central side flower, and florets are usually degenerated. Male inflorescences usually have three to five florets, consisting of one central floret and two to four side florets, with three florets being the most common. When the male inflorescence (usually located at the base of the current shoots) is stunted, a structure of one to two small flowers can be seen, and sometimes a structure of seven small flowers can be seen. Complete flowers have a similar inflorescence structure to male flowers, and the pistil of complete flowers is normally fertile. Two male sex-linked molecular markers, the DISx-AF4S and *OGI* loci, were validated in a relatively large-scale analysis of *Diospyros* L. germplasms, and the accuracy of the two SCAR (sequence-characterized amplified region) markers was nearly over 90%; they may be used to screen the sex of persimmon at the early development stage [53].

Male germplasm resources are very limited. Huang et al. found that the seven androecious persimmon germplasms were in different clustering branches, which may have two or more evolutionary origins, most of which were mutated from 'Luotian Tianshi' and 'androecious 11' may have evolved from the PCNA originally from China [54]. The androecious genotype distributed in the Dabie Mountains of Hubei Province produces a

large amount of pollen with a high germination rate, and it has the same flowering period as the main cultivars, with strong stigma compatibility; some of the androecious genotypes also have the RO2 molecular marker [55,56]. Moreover, some C-PCNA varieties, such as 'Eshi 1' and 'Luotian Tianshi', can produce male flowers [53]; therefore, these C-PCNA and androecious genotypes have the potential to be breeding parents of PCNA.

### 4.2. Artificial Pollination of Chinese Persimmon

Artificial pollination is a technique used to transfer plant pollen to the stigma by an artificial method to improve the fruit setting rate, which is a necessary means for plant hybrid breeding. Persimmon can be parthenocarpic, but sexual hybridization is still the main way to improve PCNA varieties. Sexual hybridization through artificial pollination can generate more genetic variation and provide rich resources for the breeding of excellent varieties. PCNA can naturally lose astringency on the tree and can be eaten directly after ripening, which is the main objective of genetic improvement and industrialization development. Persimmon genetic improvement in Japan began in the 1930s, and $F_1$ seeds are used in conventional crossbreeding. Through two selections, it takes 12 to 19 years to breed new sweet persimmon varieties.

The breeding of PCNA persimmon in China started late, but the development has proceeded quickly. At present, China is the global leader in the census and collection of germplasm resources, resource evaluation, breeding theory, methods, and technical practices. However, due to the difficulty of selecting the appropriate male parent, there is a large gap in the process of transferring excellent genetic genes to the offspring, and the seed formation ability is quite different among different varieties; therefore, it is difficult to conduct conventional cross-breeding. Conventional sexual hybridization breeding requires a long period of time. It usually takes 5–8 years for hybrid seedlings to bloom and bear fruits, and a large number of seedlings are needed. It takes at least 3000 progeny populations to breed excellent individual plants.

### 4.3. Embryo Rescue Technology in Chinese Persimmon

Embryo rescue is a technical way to solve the problem of the early abortion of hybrid embryos, and it is an important technology in applications involving early maturity, seedlessness, triploid breeding, the acquisition of distant hybrids, and the shortening of the breeding cycle [57]. The immature embryo inoculation period is one of the key factors that affect the success of embryo rescue. Leng et al. reported that the best time for embryo culture of 'Mopanshi' is 60–80 days after pollination [58]. Xu et al. demonstrated that the germination time of immature embryo culture from the heart-shaped stage to the cotyledon-shaped stage (50–60 days) was short, the germination rate reached 100%, and the seedling rate was 86.3–92%, which was better than that of other inoculation periods. Embryo rescue was also conducted on the young embryos of 'Luotian Tianshi' and 'Mopanshi' 50–60 days after flowering, and ideal results were obtained [59].

Moreover, the application of the embryo rescue technique in fruit trees with nucellar embryos can effectively overcome the interference of nucellar embryos and improve the success rate of obtaining hybrid offspring of polyembryonic plants [60]. In the study of the hybrid embryo culture of apricot and peach, the seed germination rate and seedling formation rate of fruit increased, and the breeding cycle was shortened by 1–2 years [61]. The embryo rescue technique can effectively avoid the sterilization of unreduced gametes in polyploid persimmon breeding. The use of embryo culture can not only effectively generate new persimmon germplasms, but also shorten the breeding period when combined with shoot grafting.

*4.4. RO2 Marker-Assisted Early Selection of the Non-Astringency/Astringency Trait in Persimmon*

The standard methods to distinguish between PCNA and non-PCNA types are the measurement of the soluble tannin content and the observation of the tannin cell size in the mature fruit. As both of these tests require fruit to be produced, it would be highly convenient to replace them with a genetic marker that could be applied to the plants at the very early stage of development. An effective SCAR marker named RO2 was first established to identify the trait of the Chinese PCNA type by Ikegami [62], derived using bulk segregant analysis and the AFLP (amplified fragment length polymorphism) technique. The RO2 marker was found in the C-PCNA cultivars 'Luotian Tianshi' and 'Baogai Tianshi' but not in any J-PCNA or non-PCNA types.

Then, the informativeness of RO2 was tested in all Chinese PCNA types, two F1 populations bred from crosses between a non-PCNA maternal parent ('Huashi 1') and either a Chinese PCNA type ('Luotian Tianshi') or an androecious male ('androecious male 3') parent. As a result, a total of 10 C-PCNA and 6 androecious genotypes were tested for the RO2 marker, and the segregation behaviour of RO2 in the $F_1$ populations bred from the two crosses showed a 1:1 ratio [63]. Moreover, the hybrid $F_1$ generations 'H1-36' [64] and 'H8-2' [37], from the crosses between the non-PCNA maternal parent 'Huashi 1' and the PCNA 'Luotian Tianshi' or androecious 8 male parent, were identified as the PCNA types using the RO2 marker and a comprehensive evaluation of fruit quality.

## 5. Genetic Transformation in Chinese Persimmon

*5.1. The Establishment of the Persimmon Regeneration System*

The establishment of the persimmon regeneration system has great significance for the rapid propagation of persimmon aseptic seedlings, shortening the juvenile period of persimmon, and obtaining transgenic plants. Liu et al. utilized stems without internodes taken from 'Mopanshi' tube seedlings as explants and demonstrated that the optimum regeneration medium for the stems without internodes of 'Mopanshi' tube seedlings was MS (1/2 N) + ZT 4.0 mg/L+ IAA 0.1 mg/L. The rate of regeneration was 83.56% and the average adventitious shoots number per explant was 2.25 [65]. Wang et al. used the stem segment of dormant buds on the branches of 'Deyangshi' as explants and found that the germination rate of the dormant buds using MS as the basic medium was 81.13%, while the germination rate of the dormant buds using MS (1/2 N) as the basic medium was 78.05%. The growth rate of the dormant buds with MS (1/2 N) as the basic medium was better than that with MS as the basic medium [66]. The tender stem segments with the buds of 'Xiaoguo Tianshi' were used as explants, which were sterilized in 75% alcohol and 0.1% HgCl$_2$ for 10 s and 8 min; then, the lowest contamination rate and browning rate were obtained, and the survival rate reached 76.67%. When the medium for the initial culture was MS (1/2 N) + 2.5 mg·L$^{-1}$ ZT + 0.1 mg·L$^{-1}$ NAA + 30 g·L$^{-1}$ sucrose and 4.5 g·L$^{-1}$ agar, the germination rate of the explants was up to 78.33%. When the medium for the subculture was MS (1/2 N) + 2.0 mg·L$^{-1}$ ZT + 0.1 mg·L$^{-1}$ NAA + 30 g·L$^{-1}$ sucrose and 4.5 g·L$^{-1}$ agar, the multiplication coefficient reached to 1.77. When the rooting medium was 1/2MS + 1.0 mg·L$^{-1}$ IBA + 30 g·L$^{-1}$ sucrose and 4.5 g·L$^{-1}$ agar, with a rooting rate of 74.73%, the average root number was 3.33, and the average root length was 2.60 cm [67]. The dormant buds were disinfected with 75% ethanol for 30 s and then disinfected with 1% NaClO for 6 min to achieve the best disinfection effect. The disinfected dormant buds were inoculated with MS (1/2 N) + 1 mg·L$^{-1}$ (ZT) + 0.1 mg·L$^{-1}$ (IAA) + 30 mg·L$^{-1}$ sucrose + 7 g·L$^{-1}$ agar + 0.6 g·L$^{-1}$ (PVP-40), and 62.56% root-free test tube seedlings were obtained. The lateral buds of the primary culture plantlets were inoculated on a medium of DKW + 1 mg·L$^{-1}$ ZT + 0.1 mg·L$^{-1}$ IAA + 1000 μmol·L$^{-1}$ betaine + 30 mg·L$^{-1}$ sucrose + 7 g·L$^{-1}$ agar + 0.6 g·L$^{-1}$ PVP-40. The average number of tillers was 1.59, the plant height was 3.30 cm, the number of leaves was 9.49 pieces·plant$^{-1}$, and the number of stem nodes was 7.40 nodes·plant$^{-1}$. The stem segments with buds were cut to 0.5 cm to continue

the proliferation culture during subculture, and the effective number of subcultures reached 10.46 per generation [68].

*5.2. Establement of the Transient Expression System*

Transient expression technology can be used to quickly verify gene function, and transient transformation system was established in vivo with *GFP* as a reporter gene by *Agrobacterium*-mediated injection infiltration in persimmon (*Diospyros kaki* Thunb.) leaves. Then, a transient ihpRNA-induced gene-silencing system based on *Agrobacterium*-mediated injection infiltration was established to evaluate the candidate genes (*phytoene desaturase*) involved in PAs biosynthesis in persimmon (*Diospyros kaki* Thunb.). Moreover, an *Agrobacterium*-mediated transient transformation system was established for gene functional analysis in persimmon (*Diospyros kaki* Thunb.) utilizing sonication followed by the vacuum infiltration of seedlings *in vitro*. To verify the function of the genes in the fruit, an easy and reliable transient guideline for further gene function verification in persimmon fruit was provided through *Agrobacterium*-mediated transformation system on fruit discs [43].

*5.3. Establement of the Stable Transformation System*

Due to the long cycle and heavy workload of conventional hybrid breeding, and hybrid offspring face difficulties, such as trait separation. Transgene technology was introduced to PCNA breeding, which has the advantages of short cycle, high efficiency, and the directional modification of individual traits. To date, there have been reports on heterozygous genes from other plant species being introduced into Japanese persimmon by stable transformation, while there are a few functional identifications of genes derived from persimmon itself by stable genetic transformation in Chinese persimmons. The regeneration rate and positive rate concerning transformation in persimmon need to be improved because of brown calli due to the high PAs content, which hinders plantlet growth [69–75]. The *Agrobacterium*-mediated ACC (1-aminocyclopropane-1-carboxylic acid) synthase gene (ACS) transformation into non-PCNA 'Mopanshi' was established, followed by 2 days of preculture time, 1.0 OD$_{600}$ of engineered bacterial suspension, 15 min of infection time, 3 days of coculture time, and 200 µmol/L AS (acetosyringone) to increase the frequency of transformation. Finally, a ACS transgenic persimmon was obtained [76]. Moreover, a two-step regeneration method was used in which the explants were cultured in a callus induction medium (MS (1/2 N)) containing 0.1 mg/L naphthalene acetic acid (NAA) and 3.0 mg/L thidiazuron (TDZ) and cultured in darkness for 10 days. Then, the leaf discs were introduced into the shoot induction medium (MS (1/2 N)) containing 1 mg/L trans-zeatin (ZT) and 0.1 mg/L indole-3-acetic acid (IAA) after 15 days of inoculation to establish the genetic transformation of *Diospyros lotus* L. As a result, the callus induction rate, average number of adventitious shoots, and regeneration rate reached 100%, 4.26, and 94.28%, respectively [77]. According to the *Agrobacterium*-mediated leaf disc infection and secondary induction system on 1/2 DKW medium + ZT (2.0 mg/L) + TDZ (0.5 mg/L) and MS (1/2 N) + ZT (2.0 mg/L) + IAA (0.1 mg/L) in non-PCNA 'Gongcheng Shuishi', the callus induction rate and adventitious bud induction rate reached 96.0% and 94.0%, respectively, and the positive rate of *DkANRi* reached 17.9%. Based on this effective protocol, transgenic lines of *DkMYB19* and *DkMYB20* with a positive rate of 28.8% were obtained [78].

## 6. Conclusions and Future Prospects

The natural deastringency of C-PCNA includes two modes, the "dilution effect" and the "coagulation effect", and two pathways, the PAs biosynthesis pathway and the acetaldehyde metabolic pathway. The regulatory network involves tannin precursor synthesis-related structural genes, tannin precursor transmembrane transport genes, tannin precursor polymerization genes, structural genes related to acetaldehyde metabolism, transcription factors, and miRNAs that regulate tannin metabolism and acetaldehyde metabolism. To date, the major structural genes related to tannin synthesis, such as

*DkANR* [79], *DkLAR* [43,80], and *DkLAC* [43]; the tannin precursor transmembrane transport genes *DkGST1* [81]/*DkGSTF1* [82], *DkMATE1/5/7* [47,83], and *DkAHA1* [84]; and the acetaldehyde metabolism genes *DkADH1*, *DkPDC2*, *DkALDH2/10*, and *DkPK1*, have been isolated and identified. Some transcription factors, such as *DkPA1*, *DkMYC1*, *DkMYB19*, *DkMYB20*, and *DkMYB21*, were verified to regulate the biosynthesis of PAs. Other transcription factors, such as *DkMYB14*, *DkMYB15*, and *DkbHLH2*, were verified to inhibit the biosynthesis of PAs, among which *DkMYB14* was also verified to promote the accumulation of acetaldehyde by upregulating the expression of *ADH* and *PDC*. Moreover, *DkWRKY3/15* was verified to regulate the expression of *DkPK1* and promote the biosynthesis of pyruvate, which results in the accumulation of acetaldehyde. In addition, miRNAs, such as miR397 and miR858, were verified to target *DkLAC2* and *DkMYB19/DkMYB20*, respectively, which are involved in the regulation of the biosynthesis of PAs, while miR143 was verified to target *DkALDH10*, which regulates the metabolism of acetaldehyde and is involved in the coagulation of soluble tannins (Figure 4).

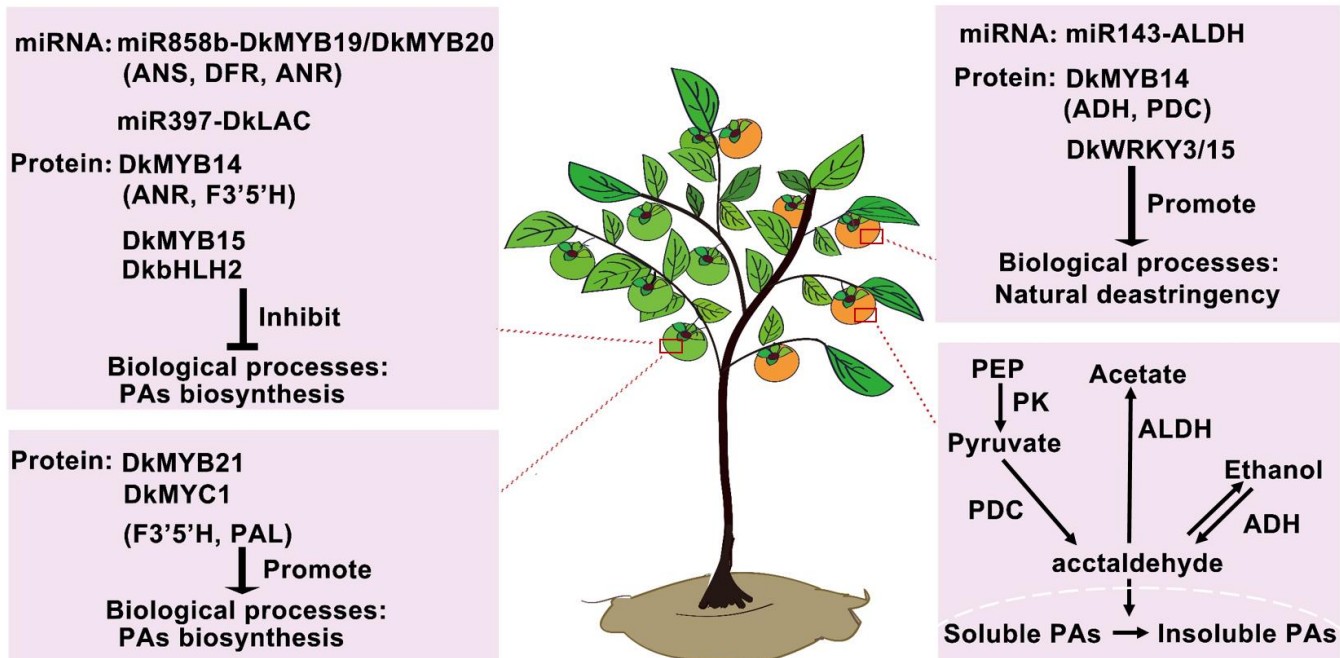

**Figure 4.** The key genes and regulatory network of tannin metabolism in C-PCNA. PEP, phosphoenolpyruvate; PK, pyruvate kinase; PDC, pyruvate decarboxylase; ADH, alcohol dehydrogenase; and ALDH, acetaldehyde dehydrogenase.

The PCNA varieties resistant to storage, no after-ripening, and are crisp and refreshing, are gradually replacing non-PCNA persimmon varieties and are favoured by the majority of consumers. Breeding new C-PCNA varieties with independent intellectual property rights plays a very important role in the high-quality development of the persimmon industry. C-PCNA and the androecious genotype are potential parents for breeding PCNA types, which could shorten the C-PCNA breeding cycle and increase the PCNA rate in the filial generation. Using the offspring of C-PCNA or the androecious genotype as materials, the key gene loci of natural deastringency may be precisely mapped utilizing BSA-seq technology. Moreover, with the publication of the persimmon genome sequence, we will screen the key genes involved in PAs biosynthesis or natural deastringency by multiomics joint analysis. Then, we will knock down the key genes involved in PAs biosynthesis using CRISPR technology and overexpress the key genes involved in acetaldehyde biosynthesis in non-PCNA to reduce the tannin accumulation ability or increase the soluble PAs coagulation ability, possibly obtaining transgenic lines of the PCNA type.

**Author Contributions:** Investigation, formal analysis, writing—original draft, and methodology, S.Y. and M.Z.; methodology, Q.Z. and Z.L.; conceptualization and resources, L.X., C.Z. and X.H.; formal analysis, supervision, and writing—review and editing, S.Y. All authors have read and agreed to the published version of the manuscript.

**Funding:** This research was supported by the Jiangxi Academy of Agricultural Sciences Ph.D. Start-up Fund (2121241) and Nanchang Comprehensive Experimental Station of National Pear Industry Technology System (CARS-28-36).

**Data Availability Statement:** Data are contained within the article.

**Acknowledgments:** The authors thank Pingxian Zhang (from the Agricultural Genomics Institute at Shenzhen, Chinese Academy of Agricultural Sciences, China) and Fatima Zaman (National Key Laboratory for Germplasm Innovation & Utilization of Horticultural Crops, College of Horticulture and Forestry Sciences, Huazhong Agricultural University, Wuhan 430070, China) for their support and improvement of this manuscript.

**Conflicts of Interest:** The authors declare no conflict of interest.

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
