# Peer review of "Recent Advances in Natural Deastringency and Genetic Improvement of Chinese PCNA Persimmon (Diospyros kaki)"

_horticulturae, doi:10.3390/horticulturae9121273_

Round 1

Reviewer 1 Report

Comments and Suggestions for Authors

The work is a review that compiles the different advances related to astringency in persimmony and the work to reduce this characteristic. Although these characteristics could be unique in each type of fruit,

oo related references from Harry Klee's group at the University of Florida are observed, for example: A chemical genetic roadmap to

improved tomato flavor. Science. January 26, 2017

Author Response

Response to Reviewer 1 Comments

Thank you very much for your review and suggestions on my manuscript. We have supplemented and modified the content of the manuscript and polished the English grammar. Thank you very much.

Reviewer 2 Report

Comments and Suggestions for Authors

The review covers all important topics for genetic improvement of natural deastringency in Persimmon. However, the over all structure of the article is a bit confusing to general readers and too much information is given without previous context. The authors need to remove redundant information and make it more concise with focus only on the trait and its genetic or natural improvement based on previous studies.

The breeding section needs to be revised, it lacks clarity and also not much information is provided. Authors should focus on breeding history of Persimmon with regard to deastringency how it evolved over time and what are the current breeding strategies, the authors should clearly provide the results of previous studies and whether the improvement was sustainable or not. Further, please see comments in the attached file.

It would also be good idea to add gene-editing section, whether any gene editing was done prior? what is the scope? can crispr-cas9 gene mediated development can be a good idea for this particular trait?

Comments on the Quality of English Language

The manuscript needs to be proofread for english language. 

Author Response

Response to Reviewer 2 Comments

Comments and Suggestions for Authors

  1. The review covers all important topics for genetic improvement of natural deastringency in Persimmon. However, the over all structure of the article is a bit confusing to general readers and too much information is given without previous context. The authors need to remove redundant information and make it more concise with focus only on the trait and its genetic or natural improvement based on previous studies.

Response: Thanks for your suggestion. This paper focuses on tannin metabolism and genetic breeding of sweet persimmon in China, including the main research progress of sweet persimmon in recent years. Redundant information and unclear descriptions in this article have been removed and modified.

  1. The breeding section needs to be revised, it lacks clarity and also not much information is provided. Authors should focus on breeding history of Persimmon with regard to deastringency how it evolved over time and what are the current breeding strategies, the authors should clearly provide the results of previous studies and whether the improvement was sustainable or not. Further, please see comments in the attached file.

Response: Thanks for your suggestion. In this paper, we have modified some contents of conventional cross breeding, mainly introduced the main strategies and difficulties of sweet persimmon breeding at present.

  1. It would also be good idea to add gene-editing section, whether any gene editing was done prior? what is the scope? can crispr-cas9 gene mediated development can be a good idea for this particular trait?

Response: Thanks for your suggestion. At present, CRISPR technology has not been reported in persimmon, but it has very promising applications in gene function verification and genetic improvement, so we hope to introduce this technology in future research work on persimmon.

According to the comments in the attached file, we have corrected follow the order.

  1. Response: Thanks for your suggestion. I have corrected the ‘can’t fully natural deastringency’ to ‘can not fully natural deastringency’.

  1. Response: Thanks for your suggestion. I have deleted the sentence ‘and can make raw leather into leather’

  1. Response: Thanks for your suggestion. I have rephrased the sentence ‘such as proanthocyanidins in feed plants can prevent the erosion of seeds by fungi’ to ‘that play an important role in protecting plants, such as proanthocyanidins can prevent the erosion of seeds by fungi in feed plants’.

  1. Response: Thanks for your suggestion. I have corrected the ward ‘disease’ to ‘diseases’.

  1. Is it daily products or daily chemical products? please take a look and rephrase the sentence to make it clear

Response: Thanks for your suggestion. It is the daily products. I have rephrased the sentence ‘persimmon tannin is also widely used as a raw material or additive for daily or chemical products and health care products’ to ‘persimmon tannin is also widely used as a raw material or additive for daily products, chemical products and health care products’.

  1. Response: Thanks for your suggestion. I have replaced the ward ‘consistency’ to ‘similarity’.

  1. discovered or developed? it seems it was not just one cultivar, please confirm

Response: Thanks for your suggestion. ‘Luotian Tianshi’ is the first discovered sweet persimmon cultivar in China.

  1. 3.2 can be joined with 3.1 and be one complete heading

Response: Thanks for your suggestion. I have merged the title 3.2 and 3.1 ‘Acetaldehyde metabolic is the essential pathway involve in natural deastringency in C-PCNA’.

  1. The 4.2 looks like a material and method section of a manuscript instead of a review article. The authors should focus on the findings and products of the previous literature using artificial pollination, its history and constraints as well not just write the method of how to do artificail pollination

Response: Thanks for your suggestion. I have rewroted this section ‘Artificial pollination is a technique used to transfer plant pollen to the stigma by an artificial method to improve the fruit setting rate, which is a necessary means for selecting parents for plant hybrid breeding. Persimmon can be parthenocarpic, but sexual hybridization is still the main way to improve PCNA varieties. Sexual hybridization through artificial pollination can generate more genetic variation and provide rich resources for the breeding of excellent varieties. The PCNA can loss the astringency natural on the tree and can be eaten directly after ripening, which is the main object of genetic improvement and industrialization development. Persimmon genetic improvement in Japan began in the 1930s, and the F1 seeds are used in conventional crossbreeding. Through two selections, it takes 12 to 19 years to breed new sweet persimmon varieties. 

The breeding of PCNA persimmon in China started late, but the development has proceeded quickly. Currently, China is the global leader in the census and collection of germplasm resources, resource evaluation, breeding theory, method and technical practices. However, due to the difficulty of selecting the appropriate male parent, there is large gap in the process of transferring excellent genetic genes to the offspring, and the seed formation ability is quite different among different varieties, so it is difficult to carry out conventional cross breeding. Conventional sexual hybridization breeding requires a long period of time. It usually takes 5-8 years for hybrid seedlings to bloom and bear fruits, and a large number of seedlings are needed. It takes at least 3000 progeny populations to breed excellent individual plants.’

  1. Response: Thanks for your suggestion. I have rewroted this section ‘Embryo rescue is a technical way to solve the problem of early abortion of hybrid embryos, and it is an important technology in applications involving early maturity, seedlessness, triploid breeding, acquisition of distant hybrids and shortening of the breeding cycle [72]. The immature embryo inoculation period is one of the key factors affecting the success of embryo rescue. Leng et al. reported that the best time for embryo culture of Mopan persimmon is 60-80 days after pollination [73]. Xu et al. demonstrated that the germination time of immature embryo culture from the heart-shaped stage to the cotyledon-shaped stage (50-60 days) was short, the germination rate reached 100%, and the seedling rate was 86.3% -92%, which was better than that of other inoculation periods. Embryo rescue was also carried out on the young embryos of ‘Luotian Tianshi’ and ‘Mopanshi’ 50-60 days after flowering, and ideal results were obtained [74].

Moreover, the application of embryo rescue technique in fruit trees with nucellar embryos can effectively overcome the interference of nucellar embryos and improve the success rate of obtaining hybrid offspring of polyembryonic plants [75]. In the study of hybrid embryo culture of apricot and peach, the seed germination rate and seedling formation rate of fruit were increased and the breeding cycle was shortened by 1-2 years [76]. The embryo rescue technique can effectively avoid the sterilization of unreduced gametes in polyploid persimmon breeding. The use of embryo culture can not only effectively generate new persimmon germplasm but also shorten the breeding period when combined with shoots grafting.’

  1. This whole paragraph needs revision, it looks like it contains incomplete sentences. the authors need to give brief but clear results of the previous stuides, for example in the first sentence; RO2 was tested in all Chinese PCNA and two F1 populations bred from cross between a non-PCNA and either a Chinese PCNA or androcious male.

the next sentence continues with a new information which seems to a reader like another 10 PCNA and 6 androecious genotypes were tested however even their results were not mentioned. all this need to be clear and with results.

Response: Thanks for your suggestion. I have revised this sentence ‘the informativeness of RO2 was tested in all Chinese PCNA types and two F1 populations bred from crosses between a non-PCNA maternal parent (‘Huashi 1’) and either a Chinese PCNA type (‘Luotian Tianshi’) or an androecious male (‘androecious male 3’) parent, a total of 10 C-PCNA and 6 androecious genotypes had been tested the RO2 marker.’

Reviewer 3 Report

Comments and Suggestions for Authors

The manuscript provides a thorough overview of persimmon fruit and its astringency caused by proanthocyanidins (PAs). It emphasizes the unique Chinese pollination constant and non-astringent (C-PCNA) variety, controlled by a single dominant locus. The review delves into the "dilution effect" and "coagulation effect" in astringency removal and the use of molecular markers for trait testing in hybrid offspring, efficiently shortening the breeding cycle. The prospect of creating a genetic transformation system for persimmons to accelerate transgene breeding is promising. This comprehensive review underscores the complexity of astringency metabolism and its importance in persimmon breeding.

Major comments:

1.     Adde references in table 1.

2.     In Line 117, the authors intend to compare PCNA and non-PCNA, but in Line 118, J-PCNA and C-PCNA were compared as an example, which is confused.

3.     In Lines 170-176, Figure 4 is referenced for these descriptions. However, I couldn't find any of these enzymes described in Figure 4.

4.     In line 274, it would be beneficial to provide a brief introduction to Artificial Pollination before delving into the technique details.

5.     The presence of numerous lengthy sentences, along with some incorrect connections between sentences, makes the text challenging to read.

6.     The genome sequences and the development of CRISPR technology are crucial for genome editing in Persimmon. However, these topics are only briefly mentioned in the conclusions. It would be beneficial to provide more detailed descriptions and references of these aspects.

Minor comments:

1.     Line 34, “(Figure),” to “(Figure).”

2.     Grammatical errors in Line 64-65, Line 68-70, Line 126-129, Line 206-210, Line 251-252, Line 329-332.

3.     Line 136, “weight of tannin was 1.38 ×104.” Please add a unit for the value.

4.     In line 194, "two process" should be corrected to "two processes." Please review the manuscript for similar errors and make necessary corrections.

5.     Line 196 “so the struture gene involve…”, I believe it should be “so the structure gene involved…”

6.     Line 203, change “349 residues” to “349 amino acid residues”

7.     In line 220, "bHLH" should be explained, as it's the first occurrence of this term and it should be italic as a gene name. Same with SCAR in line 259, RO2 in line 303, AFLP in line 305, ACC in line 366, AS in line 369.

8.     Line 252, change “Male floral sequences” to “Male inflorescence”.

9.     Line 262, change “Male germplasm resources are very limited,” to “Male germplasm resources are very limited.”

Comments on the Quality of English Language

Most of the text is written clearly. However, there are many lengthy sentences with incorrect connections, which can make the reading challenging.

Author Response

Response to Reviewer 3 Comments

Comments and Suggestions for Authors

Major comments:

(1) Adde references in table 1.

Response: Thanks for your suggestion. We have added the references in table 1.

(2) In Line 117, the authors intend to compare PCNA and non-PCNA, but in Line 118, J-PCNA and C-PCNA were compared as an example, which is confused.

Response: Thanks for your suggestion. We have modified the sentence that “The developmental dynamics of tannin cells in C-PCNA and J-PCNA are not the same; for example, except for the coagulated browning tannin cells, there are a few dark reddish brown condensed tannin cells in J-PCNA but not in C-PCNA”.

(3) In Lines 170-176, Figure 4 is referenced for these descriptions. However, I couldn't find any of these enzymes described in Figure 4.

Response: Thanks for your suggestion. The figure 4 insert position error has been corrected.

(4) In line 274, it would be beneficial to provide a brief introduction to artificial pollination before delving into the technique details.

Response: Thanks for your suggestion. We have added a brief introduction to artificial pollination before delving into the technique details, “Artificial pollination is a technical measure to transfer plant pollen to stigma by artificial method to improve fruit setting rate, that is a necessary means to select parents for plant hybrid breeding”.

(5) The presence of numerous lengthy sentences, along with some incorrect connections between sentences, makes the text challenging to read.

Response: Thanks for your suggestion. We made changes to the links and representations of long sentences in the manuscript.

(6) The genome sequences and the development of CRISPR technology are crucial for genome editing in Persimmon. However, these topics are only briefly mentioned in the conclusions. It would be beneficial to provide more detailed descriptions and references of these aspects.

Response: Thanks for your suggestion. At present, CRISPR technology has not been reported in persimmon, so we hope to introduce this technology in future research work on persimmon.

Major comments:

  1. Line 34, “(Figure),” to “(Figure).”

Response: Thanks for your suggestion. We have modified.

  1. Grammatical errors in Line 64-65, Line 68-70, Line 126-129, Line 206-210, Line 251-252, Line 329-332.

Response: Thanks for your suggestion. We have modified, “Proanthocyanidins play an important role in protecting plants, such as proanthocya-nidins can prevent the erosion of seeds by fungi in feed plants”, “persimmon tannin is also widely used as a raw material or additive for daily products or chemical products and health care products”, “In 1923, Komatsu and Mutsunami first extracted the astringent phenolic substance from the unripe astringent persimmon juice, carried out research on its molecular struc-ture, and initially defined the molecular formula of persimmon tannin as C14H20O9, further studies have shown that persimmon tannin may contain gallic acid and m-benzene”, “A DkLAC1 gene was isolated by the homology-based clone method, the predicted protein product of this gene showed that the DkLAC1 is a plant laccase which is phylogenetically related to the known enzyme AtLAC15 involved in the polymerization of PAs, and it is predicted potentially involved in PAs polymerization in C-PCNA during normal ripening”, “The tender stem segments with buds of ‘Xiaoguo Tianshi’ were used as explants, that were sterilized in 75% alcohol and 0.1% HgCl2 for 10 s and 8 min, then it had the lowest contamination rate and browning rate, and the survival rate reached to 76.67%”.

  1. Line 136, “weight of tannin was 1.38 ×104.” Please add a unit for the value.

Response: Thanks for your suggestion. We have add a unit for the value “Matsuo and Ito (1978) analyzed the tannin structure of pollination-variant astringent 'Hiratanenashi' fruit and found that the molecular weight of tannin was 1.38 ×104 Da.”

  1. In line 194, "two process" should be corrected to "two processes." Please review the manuscript for similar errors and make necessary corrections.

Response: Thanks for your suggestion. We have corrected the “two process” to “two processes”

  1. Line 196 “so the struture gene involve…”, I believe it should be “so the structure gene involved…”

Response: Thanks for your suggestion. We have corrected the “struture” to “structure”.

  1. Line 203, change “349 residues” to “349 amino acid residues”

Response: Thanks for your suggestion. We have changed the “349 residues” to “349 amino acid residues”.

  1. In line 220, "bHLH" should be explained, as it's the first occurrence of this term and it should be italic as a gene name. Same with SCAR in line 259, RO2 in line 303, AFLP in line 305, ACC in line 366, AS in line 369.

Response: Thanks for your suggestion. We have explained the abbreviated vocabulary, such as “bHLH (basic helix-loop-helix)”, SCAR (sequence-characterized amplified region), AFLP (amplified fragment length polymorphism), ACC (1-aminocyclopropane-1-carboxylic acid), AS (acetosyringone).

  1. Line 252, change “Male floral sequences” to “Male inflorescence”.

Response: Thanks for your suggestion. We have changed “Male floral sequences” to “Male inflorescence”.

  1. Line 262, change “Male germplasm resources are very limited,” to “Male germplasm resources are very limited.”

Response: Thanks for your suggestion. We have changed “Male germplasm resources are very limited,” to “Male germplasm resources are very limited.”

Round 2

Reviewer 2 Report

Comments and Suggestions for Authors

The authors have revised the manuscript as per the comments.

Author Response

Response to Reviewer Comments

Thanks a lot for comments about our manuscript.

We have modified the Abstract and the languages of the manuscript.

The revisions to the manuscript have been highlighted in the manuscript named ‘horticulturae-2650645 11.23 with marks’.

Thank you very much.

Reviewer 3 Report

Comments and Suggestions for Authors

The revisions made to the manuscript have significantly enhanced its quality. I have only a few suggestions outlined below. Some sentence are still quite lengthy and could benefit from being split into smaller, clearer sentences. For example, in line 128 there's a lack of conjunction or punctuation after "C14H20O9," which makes the sentence structure unclear. And lack of clear separation or punctuation between different clauses in lines 199-203.

Comments on the Quality of English Language

Some sentence are still quite lengthy and could benefit from being split into smaller, clearer sentences.

Author Response

(The authors gave the same response as above.)
